# Learning Is a Kan Extension

**Matthew Pugh**                                                                    *mp8g16@soton.ac.uk*
*School of Electronics and Computer Science*
*University of Southampton*
*University Road, SO17 1BJ*

**Jo Grundy**                                                                        *j.grundy@soton.ac.uk*
*School of Electronics and Computer Science*
*University of Southampton*
*University Road, SO17 1BJ*

**Corina Cirstea**                                                                   *cc2@ecs.soton.ac.uk*
*School of Electronics and Computer Science*
*University of Southampton*
*University Road, SO17 1BJ*

**Nick Harris**                                                                      *nrh@ecs.soton.ac.uk*
*School of Electronics and Computer Science*
*University of Southampton*
*University Road, SO17 1BJ*

**Reviewed on OpenReview:** *https://openreview.net/forum?id=xWKtKdeefL*

## Abstract

Previous work has demonstrated that efficient algorithms exist for computing Kan extensions and that some Kan extensions have interesting similarities to various machine learning algorithms. This paper closes the gap by proving that all error minimisation algorithms may be presented as a Kan extension. This result provides a foundation for future work to investigate the optimisation of machine learning algorithms through their presentation as Kan extensions. A corollary of this representation of error-minimising algorithms is a presentation of error from the perspective of lossy and lossless transformations of data.

## 1 Introduction

Recent work has indicated that Kan extensions have a structural similarity to many machine learning algorithms Shiebler (2022); Pugh et al. (2024). There is a tremendous amount of theory around the study of Kan extensions Perrone & Tholen (2022); Kelly (2005) and even algorithms for computing left Kan extensions efficiently Meyers et al. (2022). If the connection between Kan extensions and machine learning algorithms can be made more concrete, then it would be possible to leverage this body of work in the study of machine learning algorithms.

This paper seeks to provide a concrete connection by proving that all error minimisation problems may be presented as a left Kan extension (Thm 4.1).

A definition of error minimisation using sets and functions is lifted into the category-theoretic domain by representing it with categories and functors (Def 3.7). It is shown that error may be represented by a lax 2-functor, which associates a form of information loss to transformations between datasets (morphisms in a category) (Def 3.3).

The category-theoretic presentation of an error minimisation problem is used to show that left adjoint functors produce a global error minimiser for any input dataset (Thm 3.9). Furthermore, the error minimiser

is independent of the error, indicating that an appropriate choice of the category of datasets is sufficient to determine the global error minimisation solutions (Cor 3.10). A consequence of this result is the connection between adjoint functor theorems Porst (2024) and error minimisation problems, providing sufficient conditions to define when an optimal solution to an error minimisation problem must exist (Cor 3.12).

It is then shown that left Kan extensions are also error minimisers and that for any traditional or set-theoretic error minimisation problem, there is a 2-category whose left Kan extensions are precisely the global minimisers of the error minimisation problem (Thm 4.1).

## 2 Background

### 2.1 Categories, Adjunctions, and Kan Extensions

The definitions of categories, functors, natural transforms, adjunctions, and Kan extensions are found in all of the following resources. (Riehl, 2016; Fong & Spivak, 2018; Leinster, 2016). The definition of a 2-category is adapted from its definition as an enriched category Kelly (2005).

A category is a collection of objects and morphisms where every morphism has a domain object and codomain object. Two morphisms may be composed if the domain of one equals the codomain of the other.

**Definition 2.1** (Category). A category $C$ consists of a class of objects $Ob(C)$, and between any two objects $x, y \in Ob(C)$ a class of morphisms $C(x, y)$ such that:

- Any pair $f \in C(x, y)$ and $g \in C(y, z)$ can be composed to form $gf \in C(x, z)$.
- Composition is associative: $(hg)f = h(gf)$.
- Every object $x \in Ob(C)$ has an identity morphism $Id_x \in C(x, x)$.
- for any $f \in \mathcal{C}(x, y)$ then $fId_x = f = Id_y f$.

When clear from context, it is common to write $x \in Ob(C)$ as $x \in C$ and $f \in C(x, y)$ as $f : x \to y$. One example of a category is *Set* whose objects are sets and whose morphisms are set functions. Morphisms are often considered to be structure preserving maps. As sets have no structure by design, their morphisms are just functions. An example of a morphism between categories is a functor.

**Definition 2.2** (Functor). A functor $F : C \to D$, between categories $C$ and $D$ sends every object $x \in Ob(C)$ to $F(x) \in Ob(D)$, and every morphism $f \in C(x, y)$ to $F(f) \in D(F(x), F(y))$ such that:

- $F$ preserves composition: $F(gf) = F(g)F(f)$
- $F$ preserves identities: $F(Id_x) = Id_{F(x)}$

The product of two categories $C$ and $D$ may be written as $C \times D$. Its objects are pairs of objects from $C$ and $D$, and its morphisms are pairs of morphisms.

$$f \in C(x, y) \wedge g \in D(w, z) \implies (f, g) \in C \times D((x, w), (y, z)) \tag{1}$$

The unit of the categorical product is the category **1** which has a single object and a single morphism (which is the identity of its object). The categorical product of a category $C$ with **1** is isomorphic to $C$, meaning that there exists an invertible functor from the product into $C$. These invertible functors are referred to as the left and right unitors $l$ and $r$.

$$l : \mathbf{1} \times C \to C \tag{2}$$

$$r : C \times \mathbf{1} \to C \tag{3}$$

The left and right unitors simply drop the single object from pairs of object in $C \times \mathbf{1}$ or $\mathbf{1} \times C$. I.e. $l(*, x) = x$ and $r(x, *) = x$. The categorical product is also associative, as described by the existence of an invertible morphism $\alpha$ for triple of objects composed using the categorical product.

$$\alpha : (C \times D) \times E \to C \times (D \times E) \tag{4}$$

$\alpha$ simply rewrites nested tuples, $\alpha((x, y), z) = (x, (y, z))$.

As well as morphisms between categories it is also possible to consider the existence of morphisms between functors, called natural transforms.

**Definition 2.3** (Natural Transform). Given functors $F, G : C \to D$ between categories $C$ and $D$, a natural transformation $\eta : F \Rightarrow G$ is a family of morphisms $\eta_x : F(x) \to G(x)$ in $D$ for each object $x \in Ob(C)$, such that $G(f)\eta_x = \eta_y F(f)$ for any $f \in D(x, y)$, i.e. the following diagram commutes:

$$
\begin{array}{ccc}
F(x) & \xrightarrow{\eta_x} & G(x) \\
{\scriptstyle F(f)}\downarrow & & \downarrow{\scriptstyle G(f)} \\
F(y) & \xrightarrow[\eta_y]{} & G(y)
\end{array}
$$

A natural transform is a morphism between morphisms, referred to as a 2-morphism, whereas a morphism between objects is a 1-morphism. When the definition of a category is extended to include 2-morphisms it is referred to as a 2-category. An example of a 2-category is $Cat$, whose objects are categories, 1-morphisms are functors, and 2-morphisms are natural transforms. Given 1-morphisms $f : x \to y$ and $g : x \to y$ a two morphism $\eta$ from $f$ to $g$ may be written as $\eta : f \Rightarrow g$. Rather than hom classes a 2-category has hom-categories. It is more concise to present the definition of a 2-category using a composition functor and to present the identity morphisms with a functor $J_x : \mathbf{1} \to C(x, x)$. The functor $J_x$ selects on object of $C(x, x)$, were $J_x(*) = Id_x$. This also introduces an identity 2-morphism $Id_f$ for any 1-morphism $f : x \to y$.

**Definition 2.4** (2-category). A 2-category $C$ consists of a class of objects $Ob(C)$, and between any two objects $x, y \in Ob(C)$ a 1-category of morphisms $C(x, y)$ such that:

- For any triple of objects $x, y, z \in Ob(C)$ there is a composition functor $\circ_{x,y,z} : C(y, z) \times C(x, y) \to C(x, z)$.
- Composition is associative: $\circ_{x,y,w}(\circ_{y,z,w} \times Id_{C(x,y)}) = \circ_{x,z,w}(Id_{C(z,w)} \times \circ_{x,y,z})\alpha$.
- Every object $x \in Ob(C)$ has an identity morphism $J_x : \mathbf{1} \to C(x, x)$.
- $\circ_{x,y,y}(J_x \times C(x, y)) = l$ and $\circ_{x,y,y}(C(y, x) \times J_y) = r$

The reason for writing the definition of a 2-category using functors rather than listing the axioms of its composition of 1-morphisms and 2-morphisms is because there is a long list of axioms which are just a consequence of its composition being functorial. For example, the horizontal and vertical composition of 2-morphisms. Because 2-morphisms are 1-morphisms of their hom categories, two 2-morphisms $\eta : f \Rightarrow g$ and $\gamma : g \Rightarrow h$ may be vertically composed to form $\gamma\eta : f \Rightarrow h$. Whereas, if the 2-morphisms are side by side they may be horizontally composed via the composition functor

$$
x \Downarrow_{\eta} y \Downarrow_{\gamma} z \qquad\qquad x \Downarrow_{\gamma \circ \eta} z
$$

A 1-morphism may be composed with a 2-morphism through the process of left or right whiskering. This is simply the horizontal composition of the 2-morphism with the identity of the 1-morphism.

$$
x \xrightarrow{f} y \Downarrow_{\gamma} z \qquad x \underset{\gamma \circ Id_f}{\Downarrow} z \qquad x \underset{\gamma \cdot f}{\Downarrow} z
$$

$$
x \Downarrow_{\eta} y \xrightarrow{g} z \qquad x \underset{Id_g \circ \eta}{\Downarrow} z \qquad x \underset{g \cdot \eta}{\Downarrow} z
$$

The results of this work concern the properties of adjunctions and left Kan extensions as error minimisers. Both of these constructions may be presented in any 2-category, but the definition of an adjunction specific to adjoint functors will be of more use. A loose intuition of an adjunction between two functors is that each adjoint functor serves as an approximate or pseudo inverse for the other.

**Definition 2.5** (Adjoint Functors (triangle)). Given two functors $L : C \to D$ and $R : D \to C$, $L$ is left adjoint to $R$, and $R$ is right adjoint to $L$ (written $L \dashv R$) if and only if there exists a natural transforms $\eta : Id_C \Rightarrow RL$, called the adjunction unit, and a natural transform $\epsilon : LR \Rightarrow Id_D$, called the adjunction

counit, which given any $f : c \to R(d)$ in $C$ or $g : L(c) \to d$ in $D$ there exists $\tilde{f} : L(c) \to d$ in $D$ or $\tilde{g} : c \to R(d)$ in $C$ which are unique such that they satisfy the following commutative diagrams (triangle identities).

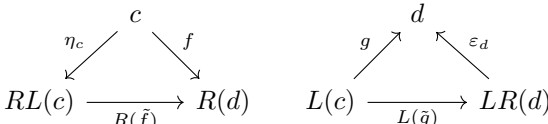

Where $\tilde{f}$ is the adjunct of $f$ constructed via $\tilde{f} := \epsilon_d L(f)$. and $\tilde{g}$ is the adjunction of $g$ constructed via $\tilde{g} := R(g)\eta_c$

**Definition 2.6** (Left Kan Extension (local))**.** Given 1-morphisms $K : C \to E$, $G : C \to D$, a left Kan extension of $K$ along $G$ is a 1-morphism $Lan_G K : D \to E$ together with a 2-morphism $\eta : K \Rightarrow (Lan_G K)G$ such that for any other such pair $(H : D \to E, \gamma : K \Rightarrow HG)$, There exists a 2-morphism $\alpha : Lan_G K \Rightarrow H$ such that $\gamma = (\alpha \cdot G)\eta$.

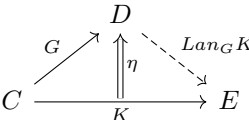

## 2.2 Monoids, Preorders, and Lax 2-Functors

The categorical description of error presented in this paper is defined using a monoidal preorder. Though this structure can be described without the use of category theory, it is presented as a kind of 2-category so that it may interact with other categorical components. Examples of the categorical definitions of otherwise describable objects are that of the Monoid and the Preorder.

**Definition 2.7** (Monoid)**.** A monoid $C$ is a category with a single object. $Ob(C) = \{*\}$.

**Definition 2.8** (Preorder)**.** A preorder $C$ is a category with at most one morphism between any two objects.

$$\forall x, y \in C(f, g \in C(x,y) \implies f = g)$$

*Remark* 2.9. The standard definition of a preorder as a transitive and reflexive relation can be recovered by taking $x \leq y$ if and only if there exists a morphism $f : x \to y$.

Though the definition of error presented later does not necessarily use the real numbers, it does require that whatever order structure is used to compare errors has a bottom or least quantity of error.

**Definition 2.10** (Bottom Element)**.** Given a preorder $P$, an element $\perp \in P$ is a bottom element of $P$ if for all $x \in P$, $\perp \leq x$.

Understanding how a monoid and a preorder may be defined from a categorical perspective makes the interpretation of the definition of a monoidal preorder more apparent.

**Definition 2.11** (Monoidal Preorder)**.** A single object 2-category with at most one 2-morphism between any pair of 1-morphisms.

Johnson & Yau (2020)

**Definition 2.12** (Lax 2-Functor between 2-categories)**.** A Lax 2-functor $F : C \to D$, sends every object $x \in Ob(C)$ to $F(x) \in Ob(D)$, it has component functors $F_{xy} : C(x,y) \to D(F(x), F(y))$ and the following natural transforms:

$$\phi : \circ_{F(x),F(y),F(z)}(F_{y,z} \times F_{x,y}) \Rightarrow F_{x,z} \circ_{x,y,z} \tag{5}$$

$$\psi : J_{F(x)} \Rightarrow F J_x \tag{6}$$

Which for all $f \in C(w,x)$, $g \in C(x,y)$, and $h \in C(y,z)$ satisfy the following constraints.

- $\phi_{h,gf}(Id_{F(h)} \circ \phi_{g,f}) = \phi_{hg,f}(\phi_{h,g} \circ Id_{F(f)})$
- $\phi_{Id_x,f}(\psi_x \circ Id_{F(f)}) = Id_{F(f)}$
- $\phi_{f,Id_w}(Id_{F(f)} \circ \psi_w)$

For the purposes of this paper, the relevant consequence of the definition of a lax 2-functor is, due to the natural transforms $\phi$ there exists a 2-morphism $\phi_{g,f} : F(g)F(f) \Rightarrow F(gf)$ in $D$ for any composable morphisms $f$ and $g$ in $C$. When the codomain of the Lax 2-functor is a monoidal preorder, the existence of a 2-morphism in the codomain can be reframed as a statement about the ordering of 1-morphisms.

**Proposition 2.13.** *Given a monoidal preorder $S$ and a lax functor $F : P \to S$ then for composable morphisms $f$ and $g$ in $P$.*

$$F(g)F(f) \leq F(gf)$$

## 3 Error

Error minimisation attempts to achieve a particular output in one space $d \in D$ of a given mapping $Inf : M \to D$ by selecting an appropriate input $m \in M$. To compare different choices of $m$ there is a bivariate function into the non negative real numbers $Err : D \times D \to \mathbb{R}_+$ which allows some measurement of difference between the actual output $Inf(m)$ and the desired output $d$. Such a problem may be codified with sets and functions.

**Definition 3.1** (Set Theoretic Error Minimisation Problem)**.**

$$
\begin{aligned}
Given \quad & M, \ D \in Set \\
& Inf \in Set(M, \ D) \\
& Err \in Set(D \times D, \ \mathbb{R}_+) \\
& d = d' \implies Err(d, d') = 0 \\
& d \in D \\
\text{minimise} \quad & Err(d, \ Inf(m))
\end{aligned}
$$

To minimise $Err(d, \ Inf(m))$ means to select a global error minimiser with respect to $d$.

**Definition 3.2** (Global Error Minimiser)**.** Given an error minimisation problem (Def 3.1), $x \in M$ is a global error minimiser with respect to $d \in D$ if for any $m \in M$ then $Err(d, Inf(x)) \leq Err(d, Inf(m))$.

The choice to call the mapping between input and output $Inf$ is in direct reference to the notion of model inference, where a machine learning model is used to predict an output given an input. Model inference is often referred to in the context of individual inputs vs outputs. The $Inf$ function maps a particular parametrisation of a machine learning model onto the dataset it produces when allowed to produce inference over the entire set of training inputs. From this perspective, $M$ represents the set of all choices of parameters for the machine learning model and $D$ is the set of all datasets that one may try to train against.

In order to apply the category theoretic constructions of adjunctions and Kan extensions to this definition, it needs to be lifted into a description using categories and functors. This is easily done with respect to $Inf$ by the statement that $M$ and $D$ should be categories and $Inf : M \to D$ a functor. The next question is how to categorify the notion of error. Morphisms are commonly thought of as structure preserving transformations. In the context of $D$ this would suggest that the morphisms are transformations between datasets. Data transformations, functions, or programs can only lose information. They cannot add information that wasn't previously there. The better one dataset represents the information of another, the less information loss a mapping between them may experience. This would indicate that error may be associated to a category by assigning each morphism some quantity of error that represents its information loss. The values which represent error require some order structure so they can be compared and should reflect how error composes as morphisms compose. This suggests that the values of error may be represented by a monoidal preorder (Def 2.11).

**Definition 3.3** ($S$ Flavoured Error)**.** Given a monoidal preorder $S$, where $Id_*$ is the bottom element, then $S$ flavoured error on $D$ is a lax 2-functor $Err : D \to S$.

To make use of the structure of $D$, the choice of error should respect the information that $D$ contains. Namely, the composition of morphisms. A mapping of morphisms to morphisms which respects their composition would usually be indicative of a functor. However, though it would work, an error functor would be an excessively restrictive constraint. In practice, the information loss of the composite of two processes cannot usually be represented by the composition of the information loss of each of the processes individually. Two processes may lose the same portion of information, so their composite loss is not much worse than their individual losses. In contrast, the lossy-ness of a different pair of processes may affect entirely different portions of the information content, so their composite information loss would be much larger than their individual losses. It is much easier to represent the information loss of the composite of two morphisms via an inequality, which can be done using a lax 2-functor, encoding the relationship that the error of a composition of morphisms must be greater than or equal to some composition of their errors.

$$Err(g)Err(f) \leq Err(gf) \tag{7}$$

One example of a suitable monoidal preorder would be the single object category $\mathbb{R}_\wedge$ whose morphisms are the non-negative real numbers composed by taking the maximum and ordered by the standard ordering on the reals. Imagine the case of the objects of $D$ being data streams with morphisms being functions which map one data stream to another. In this case, the system is well described by an $\mathbb{R}_\wedge$ flavoured error on $D$. The functions between data streams have some associated information loss. If one is considering the error to be measured purely by the lost information and not just some invertible scrambling, then it wouldn't be possible to undo the error. So the error of those two functions composed together must always be greater than the maximum of the two errors.

$$Max(Err(g), Err(f)) \leq Err(gf) \tag{8}$$

From a choice of $S$ flavoured error, it is possible to recover an ordering of error associated with pairs of objects of $D$ by looking at the best case scenario, the least errorful morphism.

**Definition 3.4** (Error Comparison)**.** Given $S$ flavoured error on $D$. For objects $x, y, z, w \in D$ then $Err(x, y) \leq_S Err(z, w)$ if and only if, for any $f : z \to w$ there exists a $g : x \to y$ and $\sigma : Err(g) \Rightarrow Err(f)$.

*Remark* 3.5. The value $Err(x, y)$ is a notational convenience and is not an object in $S$. Instead, the important aspect of error in the traditional case is that it induces a preorder on pairs of objects. Def 3.4 is a way of inducing a preorder on pairs of objects of $D$ using $S$ flavoured error.

**Proposition 3.6.** *The error comparison of an $S$ flavoured error on $D$ defines a preorder.*

*Proof.* For any morphism $f : x \to y$ there is an identity 2-morphism $Id_{Err(f)} : Err(f) \Rightarrow Err(f)$ which implies that $Err(x, y) \leq_S Err(x, y)$, demonstrating that the relation is reflexive.

If $Err(x, y) \leq Err(w, z)$ and $Err(w, z) \leq Err(a, b)$ then for any morphism $f : a \to b$ there is a morphism $g : w \to z$ and 2-morphism $\sigma : Err(g) \Rightarrow Err(f)$. Given the existence of $g$, there must be a morphism $h : x \to y$ with associated 2-morphism $\varphi : Err(h) \Rightarrow Err(g)$, which by composition induces $\sigma\varphi : Err(h) \Rightarrow Err(f)$ for any $f$ implying that $Err(x, y) \leq_S Err(a, b)$, demonstrating that the relation is transitive. As it is both reflexive and transitive the error comparison relation is a preorder. $\square$

By combining the requirement that $Inf : M \to D$ is a functor, with a choice of $S$ flavoured error on $D$, one may produce a category-theoretic definition of an error minimisation problem.

**Definition 3.7** (Category Theoretic Error Minimisation Problem)**.**

$$
\begin{aligned}
Given \quad &M, \ D \in Cat \\
&Inf \in Cat(M, \ D) \\
&S \text{ flavoured error on } D \\
&d \in D \\
return \quad &A \text{ global error minimiser with respect to } d
\end{aligned}
$$

*Remark* 3.8. By fixing $d$, Def 3.7 may also serve as a definition for category theoretic loss minimisation.

The value of translating the set-theoretic definition into the category theoretic definition is that the existence of morphisms between models and datasets provides more information about the structure of the problem. The first observation to make about the additional information is that it constrains the choices of error to those which respect the structure of the category. From a practical perspective, this provides a novel approach to the selection of an error function given a particular error minimisation problem, if one knows the morphisms between datasets. The second observation is that because the choice of error is now dependent on the morphisms of $D$, the properties of category theoretic constructions which reference only morphisms may be translated into their consequences with respect to error. The first such consequence is that adjunctions are error minimisers.

**Theorem 3.9** (Adjunctions are Error Minimisers). *Given a category-theoretic error minimisation problem (Def 3.7) where $Inf : M \to D$ has a left adjoint $Alg : D \to M$, then for all $d \in D$, $Alg(d)$ is a global error minimiser with respect to $d$.*

*Proof.* For any $d \in D$ show that $Alg(d)$ is a global error minimiser with respect to $d$ by demonstrating that for any $m \in M$, $Err(d, InfAlg(d)) \leq_S Err(d, Inf(m))$.

If there does not exist a morphism $f : d \to Inf(m)$ then the error comparison requirement (Def 3.4) is trivially satisfied.

If there does exists a morphism $f : d \to Inf(m)$ then by the definition of an adjunction (Def 2.5) for any morphism $f : d \to Inf(m)$ there is a unique morphism $\tilde{f} : Alg(d) \to m$ such that $Inf(\tilde{f})\eta_d = f$, where $\eta : Id_D \Rightarrow InfAlg$ is the adjunction unit.

By the definition of a lax 2-functor (Def 2.12) there exists a 2-morphism.

$$\sigma : Err(Inf(\tilde{f}))Err(\eta_d) \Rightarrow Err(Inf(\tilde{f})\eta_d) = Err(f)$$

Because the identity of $S$ is the bottom element there is a 2-morphism $\varphi : Id_* \Rightarrow Err(Inf(\tilde{f}))$ which by right whiskering produces the 2-morphism

$$\varphi \cdot Err(\eta_d) : Err(\eta_d) \Rightarrow Err(Inf(\tilde{f}))Err(\eta_d)$$

Compose this with $\sigma$.

$$\sigma(\varphi \cdot Err(\eta_d)) : Err(\eta_d) \Rightarrow Err(f)$$

As this is true for any choice of $f$ this proves the error comparison.

$$Err(d, InfAlg(d)) \leq_S Err(d, Inf(m))$$

As the error comparison is true for any choice of $m$ then $Alg(d)$ is a global error minimiser with respect to $d$. $\qquad\square$

**Corollary 3.10.** *The left adjoint of $Inf$ is an error minimiser for any choice of error on $D$. If one can compute the left adjoint of $Inf$ than they may identify the global error minimiser, with respect to any $d$, without ever making a particular choice of error.*

*Remark* 3.11. If a true error minimising algorithm returns a global error minimiser for any element $d \in D$, then one could define algorithms as left adjoint to inference.

If an inverse to inference exists then it would make sense it would represent an error minimisation algorithm. Ideally, any algorithm would return a model which produced exactly the intended dataset. In the case that one is not capable of reproducing exactly the desired dataset, then a pseudo inverse to inference would be an intuitive choice. In the context of category theory, the natural choice of pseudo inverse is an adjunction, but this does not have to be an adjunction of functors. The definition of an adjunction can be generalised to any 2-category. If one believes that algorithms are left adjoint to inference, then whatever object one uses to represent the collection of models and datasets, if these objects exist in a suitable 2-category, then one can define what a global error minimising algorithm should be.

$$M \underset{Alg}{\overset{Inf}{\rightleftarrows}} D$$

**Corollary 3.12.** *If an error minimising problem is presented in the categorical form, then Theorem 3.9 shows that one may prove that a global minimiser exists by proving that a left adjoint to $Inf$ exists. This statement may be repackaged with the various adjoint functor theorems Porst (2024) to produce sufficient conditions for the existence of optimal solutions to error minimisation problems.*

While the production of sufficient conditions for the existence of global error minimisers is an incredibly powerful result, it is excessively strict to always require the existence of an adjunction to discuss global error minimisers in a category-theoretic context. Furthermore, there are many practical problems where a global error minimiser may exist for some datasets but not for others. It would be more helpful to provide a construction which exists as long as a particular dataset has a global error minimiser. Such a construction would be the left Kan extension. The application of a left Kan extension to a category theoretic error minimisation problem requires the problem to be represented as an extension triangle. This can be done with a suitable choice of 2-category.

**Proposition 3.13** (Extension Error Minimisation). *An extension triangle (see below) in a 2-category $\mathbb{T}$ with an S flavoured error on $\mathbb{T}(\delta, \tau)$ defines a category theoretic error minimisation problem (Def 3.7)*

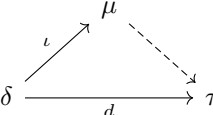

*Proof.* The 1-morphism $\iota : \delta \to \mu$ and the object $\tau$ induce the functor $\mathbb{T}(\iota, \tau) : \mathbb{T}(\mu, \tau) \to \mathbb{T}(\delta, \tau)$. The functor $\mathbb{T}(\iota, \tau)$ is defined via precomposition, sending any object $m \in \mathbb{T}(\mu, \tau)$ to $m\iota \in \mathbb{T}(\delta, \tau)$. By renaming the functor and categories as $Inf : M \to D$ it is clear that they define a category theoretic error minimisation problem. $\square$

*Remark* 3.14. It is also the case that any category-theoretic error minimisation problem may be presented as an extension in a 2-category $\mathbb{T}$ by directly defining the hom categories and composition functor of $\mathbb{T}$ to be the categories and inference functor of the error minimisation problem. Proving this is also true for the set-theoretic error minimisation problem is slightly trickier (Thm 4.1).

**Theorem 3.15** (Kan Extensions are Error Minimisers). *Given a category theoretic error minimisation problem (Def 3.7) in the form of an extension (Prop 3.13), the left Kan extension $Lan_\iota d$ is, if it exists, a global error minimiser with respect to d.*

*Proof.* For any $d \in D$ show that $Lan_\iota d$ is a global error minimiser by demonstrating that for any $m \in M$, $Err(d, Inf(Lan_\iota d)) \leq_S Err(d, Inf(m))$.

If there does not exist a morphism $f : d \to Inf(m)$ then the error comparison requirement (Def 3.4) is trivially satisfied.

If there does exist a morphism $f : d \to Inf(m)$ then this is also a 2-morphism, $f : d \Rightarrow m\iota$ (recalling that $Inf(m) = m\iota$ as described in Prop 3.13) in $\mathbb{T}$. By the definition of a left Kan extension (Def 2.6), for any 2-morphism $f : d \Rightarrow m\iota$ there exists a 2-morphism $\alpha : Lan_\iota d \Rightarrow m$ such that $f = (\alpha \cdot \iota)\eta$. These 2-morphisms in $\mathbb{T}$ correspond directly with 1-morphisms of $D$ i.e. $\eta : d \to Inf(Lan_\iota d)$ and $\alpha \cdot \iota : Inf(Lan_\iota d) \to Inf(m)$ where $Inf(Lan_\iota d) = (Lan_\iota d)\iota$.

By the definition of a lax 2-functor (Def 2.12) there exists a 2-morphism.

$$\sigma : Err(\alpha \cdot \iota)Err(\eta) \Rightarrow Err((\alpha \cdot \iota)\eta) = Err(f)$$

Because the identity of $S$ is the bottom element there is a 2-morphism $\varphi : Id_* \Rightarrow Err(\alpha \cdot \iota)$ which by right whiskering produces the 2-morphism

$$\varphi \cdot Err(\eta) : Err(\eta_d) \Rightarrow Err(\alpha \cdot \iota)Err(\eta_d)$$

Compose this with $\sigma$.

$$\sigma(\varphi \cdot Err(\eta)) : Err(\eta_d) \Rightarrow Err(f)$$

As this is true for any choice of $f$ this proves the error comparison.

$$Err(d, Inf(Lan_\iota d)) \leq_S Err(d, Inf(m))$$

As the error comparison is true for any choice of $m$ them $Lan_\iota d$ is a global error minimiser. $\square$

## 4 Universal representation

It has been shown that Kan extensions represent global error minimisers for category-theoretic error minimisation problems, but it may not be clear that this also applies to the set-theoretic error minimisation problem. It is actually possible to convert any set-theoretic error minimisation problem into a category-theoretic error minimisation problem, namely as an extension problem in a 2-category, such that the left Kan extensions of the extension problem are exactly the global error minimisers of the set-theoretic error minimisation problem.

**Theorem 4.1** (Machine Learning representation). *Given a set theoretic error minimisation problem (Def 3.1) there exists a 2-category $\mathbb{T}$ such that $M = \mathbb{T}(\mu, \tau)$, $D = \mathbb{T}(\delta, \tau)$, $Inf = \mathbb{T}(\iota, \tau)$ and an object $m \in M$ is a global error minimiser with respect to $d$ if and only if $m \cong Lan_\iota d$*

*Proof.* Construct $\mathbb{T}$ to have three objects, $\mu$, $\delta$, and $\tau$. The hom objects will be selected such that $Inf$ becomes a composition morphism, and the 2-morphisms (morphisms of the hom category) are constructed to artificially select a minimising element if it exists.

Define the following singleton categories

$$\mathbf{1} \cong \{\iota\} \cong \{Id_\mu\} \cong \{Id_\delta\} \cong \{Id_\tau\}$$

Define $\mathbf{M}$ such that $Obj(\mathbf{M}) = M$ and that for any $m, m' \in \mathbf{M}$ there is a unique morphism $\sim : m \to m'$ if and only if $Inf(m) = Inf(m')$. Define $\mathbf{D}$ to be the category whose objects are the elements of $D$. Let $U \subseteq D$ be the subset of datasets for which an error minimising model exists, and let $Alg : U \to M$ be a function which selects an error minimising model for each $d \in D$ under the constraint that if there exists an $m \in \mathbf{M}$ such that $Inf(m) = d$, then $Alg(d) \cong m$.

Define the hom sets of $\mathbf{D}$ with the following piecewise function.

$$\mathbf{D}(d, d') := \begin{cases} \{Id_d\} & d = d' \\ \{*\} & d \in U \land d' = Inf(Alg(d)) \land d \neq d' \\ \emptyset & else \end{cases}$$

Composition is defined in the obvious way. For objects $d, d', d'' \in D$ consider the form of the composition morphism.

$$\circ_{d,d',d''} : \mathbf{D}(d, d') \times \mathbf{D}(d', d'') \to \mathbf{D}(d, d'')$$

Whenever $\mathbf{D}(d, d')$ or $\mathbf{D}(d', d'')$ is empty, then the product is empty, making the composition morphism the unique map from the empty set. When both $\mathbf{D}(d, d')$ or $\mathbf{D}(d', d'')$ are non empty, they must both be singleton. Therefore the following must be true.

$$d = d' \lor (d' = Inf(Alg(d)) \land d \neq d')$$
$$d' = d'' \lor (d'' = Inf(Alg(d')) \land d' \neq d'')$$

Which may be simplified to form the following.

$$d = d' \vee d' = Inf(Alg(d))$$
$$d' = d'' \vee d'' = Inf(Alg(d'))$$

Combining these statements produces the following deduction.

$$\mathbf{D}(d, d') \times \mathbf{D}(d', d'') \cong \mathbf{1}$$
$$\implies (d = d' \vee d' = Inf(Alg(d)))$$
$$\wedge (d' = d'' \vee d'' = Inf(Alg(d')))$$
$$\implies (d = d' \wedge d' = d'')$$
$$\vee (d = d' \wedge d'' = Inf(Alg(d')))$$
$$\vee (d' = Inf(Alg(d)) \wedge d' = d'')$$
$$\vee (d' = Inf(Alg(d)) \wedge d'' = Inf(Alg(d')))$$
$$\implies (d = d'')$$
$$\vee d'' = Inf(Alg(d))$$
$$\vee (d' = Inf(Alg(d)) \wedge d'' = Inf(Alg(d')))$$

When $d' = Inf(Alg(d))$ then for $m = Alg(d)$

$$Err(Inf(m), d') = Err(Inf(Alg(d)), d') = Err(d', d') = 0$$

By the definition of $Alg$ this forces $Alg(d') \cong m = Alg(d')$, which by the construction of $\mathbf{M}$ means that $Inf(Alg(d')) = Inf(Alg(d)) = d'$. This allows the deduction to be simplified to the following implication.

$$\mathbf{D}(d, d') \times \mathbf{D}(d', d'') \cong \mathbf{1}$$
$$\implies (d = d'') \vee d'' = Inf(Alg(d))$$
$$\vee (d' = Inf(Alg(d)) \wedge d'' = Inf(Alg(d')))$$
$$\implies (d = d'') \vee d'' = Inf(Alg(d))$$
$$\vee (d' = Inf(Alg(d)) \wedge d'' = d'))$$
$$\implies (d = d'') \vee d'' = Inf(Alg(d))$$
$$\implies \mathbf{D}(d, d'') \cong \mathbf{1}$$

Making the composition morphism in this case, the unique morphism between singleton sets.

Using the above-defined categories, define the hom-categories of $\mathbb{T}$ as follows.

| $\mathbb{T}(-, -)$ | $\mu$ | $\delta$ | $\tau$ |
|---|---|---|---|
| $\mu$ | $\{Id_\mu\}$ | $\emptyset$ | $\mathbf{M}$ |
| $\delta$ | $\{\iota\}$ | $\{Id_\delta\}$ | $\mathbf{D}$ |
| $\tau$ | $\emptyset$ | $\emptyset$ | $\{Id_\tau\}$ |

The only composition morphism which is not fixed by identity laws or the empty categories is the following

$$\circ_{\delta, \mu, \tau} : \mathbb{T}(\delta, \mu) \times \mathbb{T}(\mu, \tau) \to \mathbb{T}(\delta, \tau)$$

Substituting the known hom objects, this is rewritten as.

$$\circ_{\delta, \mu, \tau} : \{\iota\} \times \mathbf{M} \to \mathbf{D}$$

Because $\{\iota\} \times \mathbf{M} \cong \mathbf{M}$, the composition morphism can be defined by the inference function which maps all morphisms of $\mathbf{M}$ to the relevant identity morphisms

$$\circ_{\delta,\mu,\tau} := Inf$$

Finally, consider the following Kan extension problem in $\mathbb{T}$.

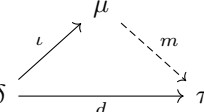

If an error minimising $m$ does not exist for the given $d$ then no Kan extension can exist as there is no morphism from $d$ into the image of $Inf : \{\iota\} \times \mathbf{M} \to \mathbf{D}$. However, if an error minimising $m$ does exist then by construction there is a morphism in $\mathbf{D}$ and consequently a 2-morphism in $\mathbb{T}$ of the form $d \Rightarrow Alg(d)\iota = Inf(Alg(d))$. For any $m$ for which there also exists a 2-morphism $d \Rightarrow m$ then as such a 2-morphism from $d$ into the image of $Inf$ is unique, then $m = Inf(Alg(d))$, which by the construction of $\mathbf{M}$ means that $Alg(d) \cong m$. This makes $Alg(d)$, when it exists, a left Kan extension in $\mathbb{T}$ □

The strategy for constructing $\mathbb{T}$ amounts to appending morphisms to $M$ and $D$ such that the Kan extension selects the minimising elements of the error functions, if they exist. The morphisms in $D$ effectively present a relation which points from any element $d \in D$ to its respective error minimiser, when it exists. The appended morphisms and sets are the minimal structure required to admit a Kan extension. Consequently, the construction of $\mathbb{T}$ acts as a minimal or free construction regarding the relevant set theoretic error minimisation problem. It is not necessarily unique as a 2-category which admits a Kan extension for a given error minimisation problem, but it will be a subcategory of any other 2-category which does admit such an extension. The intuition of this construction can be seen by considering the cases of convex optimisation and linear regression.

*Example* 4.2 (Convex optimisation). Let $M \subset \mathbb{R}^2 := D$ be a closed convex proper subset of the 2D plane, $D$. The standard euclidean distance $Err = ||x - y|| : \mathbb{R}^2 \times \mathbb{R}^2 \to \mathbb{R}$ can be used as an error function to determine the distance between two points in $D$. Because $M$ is a subset, its inclusion in $D$ may be presented as the injective function $Inf : M \rightarrowtail D$. These definitions of Inf and Err satisfy the requirements of a set theoretic error minimisation problem. Given some point $d \in D$, the error minimiser will be the unique point in $M$ which is closest to $d$. Consequently, by Thm 4.1, there exists a two category $\mathbb{T}$ such that these nearest points are Kan extensions. Figure 1 shows an example morphism in $\mathbf{D}$ when $d \neq Inf\,Alg(d) = d'$. In this instance, as $d$ is outside of $M$ it cannot be equal to its minimiser. Therefore $\mathbf{D}(d, d') = \{*\}$. Because $\mathbf{D}$ is a hom-category of $\mathbb{T}$, the 1-morphism $* : d \to d'$ in $\mathbf{D}$ is the 2-morphism in $\mathbb{T}$ required to make $Alg(d)$ a Kan extension of $d$ along $\iota$. If $d$ was selected such that $d \in M$, then $d = Inf(Alg(d)) = d'$ and $\mathbf{D}(d, d') = \{Id_d\}$. This identity morphism would then form the 2-morphism required in $\mathbb{T}$ to form the requisite Kan extension.

Having demonstrated how the nearest point within a convex set may be presented as a Kan extension, it is only a small modification to present this result in the context of linear regression.

*Example* 4.3 (Linear regression). Consider a dataset of $n$ ordered pairs $(x_i, d_i) \in \mathbb{R} \times \mathbb{R}$. The line of best fit is the affine function $f(x) = ax + b$ which minimises the root mean squared error.

$$Err(d, f) = \sqrt{\frac{1}{n} \sum_i^n (f(x_i) - d_i)^2}$$

One can consider the dataset of values to be the n-tuple of values, $d \in \mathbb{R}^n$. The set of models $M$ is the set of affine functions $f : \mathbb{R} \to \mathbb{R}$ which can be parametrised as a pair of real numbers $(a, b) \in \mathbb{R}^2 := M$. The inference function, $Inf : M \to D$, maps $M$ into $D$ by evaluating a given affine function $f$ at every point $x_i$, $Inf(f)_i = f(x_i)$. $Inf$ is a linear map from $R^2$ to $R^n$ which, in the context of linear regression, is commonly referred to as the design matrix. Using this construction of $Inf$ the error function becomes the standard euclidean distance on $D$. Analogous to the previously presented convex optimisation problem,

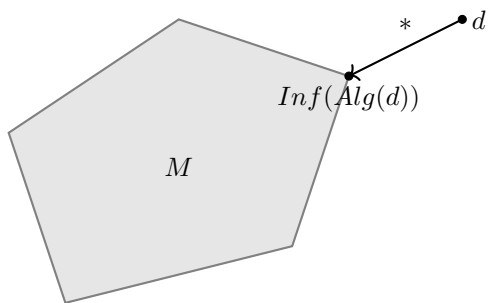

Figure 1: M is presented as a convex subset set of the plane. A morphism $*: d \to Inf(Alg(d))$ is appended between $d$ and the closest point to $d$ in $M$ such that a Kan extension of $d$ identifies $Inf(Alg(d))$ as the only possible extension.

the selection of a global minima given a particular $d$ amounts to selecting the point in the convex subset $M$ which is closest to $d$. As before, so that a Kan extension exists in $\mathbb{T}$ whenever an error minimiser exists, a morphism $*: d \to Inf(Alg(d))$ is added to $\mathbf{D}$ whenever $Inf(Alg(d))$ exists and $d \neq Inf(Alg(d))$. For the particular case of linear regression, an error minimiser exists for every choice of $d$. This means that not only can the error be presented as a Kan extension in $\mathbb{T}$, but that they also form an adjunction $Inf \vdash Alg$. As the inference mapping in this case is linear and the error function is euclidean distance, the mapping which identifies the error minimiser, Alg, is the Moore-Penrose pseudo inverse of Inf. This connection re-enforces the relationship between adjunctions, optimisation algorithms, and pseudo inverses.

The special case of linear regression presenting $Alg$ as the Moore-Penrose pseudo inverse indicates a more general aspect of the $\mathbb{T}$ construction. If one has an algorithm for Computing $Inf: M \to D$ and $Alg: U \to M$ then it is possible to query information about the 2-morphisms of $\mathbb{T}$ with a worst case time complexity of the sum of the time complexities of $Inf$ and $Alg$. This comes directly from determining the hom-set $\mathbf{D}(d, d')$, which when $d \neq d'$ and $d \in U$ is equal to the set $\{ * \,|\, Inf(Alg(d)) = d'\}$. This can be used to show that the time complexity of computing $\mathbb{T}$ is entirely dependent on the problem domain. Presume that for any algorithm which computes the function $f: X \to Y$ there is a known time complexity lower bound of $O_f$. This function may be presented as the solution to an error minimisation problem.

**Proposition 4.4** (Error Minimisation Function Representation). *For any set function $f: X \to Y$ there exists a set theoretic error minimisation problem $E_f$ (Def 3.1) with $Inf: Y \to X + Y$ such that for any point $d \in X + Y$ the point $\{f, Id_Y\}(d)$ is its global error minimiser.*

*Proof.* Construct the $Inf$ function as the canonical inclusion $Inf := i: Y \to X + Y$, and define $Err: (X + Y) \times (X + Y) \to \mathbb{R}$ as the following piecewise function.

$$Err(d, d') := \begin{cases} 0 & d = d' \vee f(d) = d' \\ 1 & else \end{cases}$$

This defines a set theoretic error minimisation function according to Def 3.1. The function $\{f, Id_Y\}: X + Y \to Y$ selects an error minimiser. This function may be defined explicitly as follows.

$$\{f, Id_Y\}(d) := \begin{cases} f(d) & d \in X \\ d & d \in Y \end{cases}$$

In the first case assume $d \in Y$.

$$Err(d, i\{f, Id_Y\}(d)) = Err(d, d) = 0$$

In the second case assume $d \in X$

$$Err(d, i\{f, Id_Y\}(d)) = Err(d, f(d)) = 0$$

In both cases, as 0 is the smallest possible value of the given error function, then $\{f, Id_Y\}(d)$ is the global error minimiser with respect to $d$. $\qquad\square$

By using Thm 4.1 to construct $\mathbb{T}$ with respect to $E_f$, as shown in Prop 4.4, one can see that the worst case time complexity for computing $\mathbf{D}(d, d')$ when $d \neq d'$ is the time complexity of $i\{f, Id_Y\}$, which in the worst case has lower bound $O_f$. This demonstrates that one can construct error minimisation problems such that querying the construction of their respective 2-category $\mathbb{T}$ must have a worst-case time complexity lower bound of $O_f$. As the function $i\{f, Id_Y\}$ also serves to compute Kan extensions, by identifying error minimisers, the computational complexity of computing Kan extensions would also be $O_f$. By selecting arbitrarily hard problems, one can see that the possible complexity of constructing $\mathbb{T}$ is unbounded. One can infer from this that any 2-category which solves $E_f$ via Kan extensions would have a worst case time complexity lower bound of $O_f$ for the computation of these Kan extensions. This can either be seen directly as these categories compute $f$ for which the time complexity is known, or as these categories contain $\mathbb{T}$ as a subcategory.

While an efficient algorithm is known, it would be unnecessary to compute $\mathbb{T}$ directly; however, the opposite problem may be susceptible to a Kan extension approach. If $f : X \to Y$ is a function which is desired to be computed but has no known efficient algorithm of computation, then by presenting this function as the solution to an error minimisation problem, it may be possible to develop methodologies for its computation via the known processes of computing Kan extensions. Particularly when these Kan extensions exist between categories with well known structures. While it should again be noted that the time complexities of these algorithms are highly dependent on the problem domain, it provides another avenue of attack by which researchers may approach the problem of algorithm design.

## 5 Conclusion

This paper has introduced an important bridge in the field of category theory for machine learning, providing a connection that allows the application of powerful category theoretic tools to old problems.

In addition to the theorems relating to the category-theoretic constructions, this methodology has also introduced insights that may impact how one views machine learning problems.

Firstly, the introduction of $S$ flavoured error supplements rigorous notions of how one might select an error function. Suggesting that error should be associated with the transformations between datasets gives an indication of how one may appropriately choose an error for a given problem. From this perspective, traditional notions such as distance, accuracy, and information loss reveal themselves as measurements of the minimal transformation necessary to convert one dataset into another.

Secondly, the independence of the left adjoint to choices of error may indicate that there are more fundamental ways of selecting a globally optimal model with respect to a dataset without referring to error. Re-framing model inference as a way of producing a dataset from a model, a mapping from a space of models to a space of datasets, allows one to think of algorithms as pseudo inverses to inference. In category theory, the natural choice of pseudo inverse is the adjunction, which is not constrained to an adjunction of functors. The definition of an adjunction can be generalised to any 2-category. Suppose it is more appropriate to represent the spaces of models and datasets as some other object, such as manifolds or measure spaces. In that case, finding an appropriate choice of 2-morphisms will indicate how to construct an algorithm as a left adjoint to inference.

$$M \underset{Alg}{\overset{Inf}{\underset{\top}{\rightleftarrows}}} D$$

Finally, the demonstration that left Kan extensions may represent any error minimisation problem provides an interesting connection between the purpose of machine learning and the presentation. It is often intuitive to think that machine learning models find patterns within a dataset, allowing it to extend the already present data. However, this is only possible if one introduces additional assumptions about the data. Assumptions such as linearity, distance, smoothness, and maximum likelihood are all examples of assumptions machine

learning models utilise to extend datasets. The nature of taking some aspect of data and extending it to a different context is precisely the structure encoded by a Kan extension, connecting intuitions about machine learning to a rigorous algebraic representation.

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
