# OpenReview forum: "Learning Is a Kan Extension"
_TMLR — Accepted by TMLR_

### Review · Reviewer_Jc1D · 2025-08-11

**Summary Of Contributions:**

In this paper, the authors prove a general result showing that any error minimization problem can be represented as a left Kan extension in a suitably defined 2-category. This builds on prior work noting that certain ML algorithms resemble Kan extensions, but goes further by showing that all error minimization algorithms admit such a representation. The paper develops the notion of “S-flavored error” via lax 2-functors, proves that adjunctions are global error minimizers (Theorem 3.9), generalizes this to left Kan extensions (Theorem 3.15), and finally shows that any set-theoretic error minimization problem can be embedded into a 2-category whose left Kan extensions are exactly the global minimizers (Theorem 4.1). This provides a clean categorical formalism that connects machine learning optimization to a large body of existing results in category theory.

**Audience:**

Yes

**Audience Explanation:**

This paper presents a substantial theoretical contribution, establishing a broad categorical unification of error minimization problems via left Kan extensions. The work will be of interest to the theory-oriented portion of TMLR’s readership, particularly those at the intersection of category theory and machine learning. That said, the paper’s impact and accessibility would be improved by adding at least one worked-through example, some discussion of computational aspects, and lighter notation where possible.

**Broader Impact Concerns:**

This is primarily a theoretical paper with no immediate societal risks beyond those of ML in general. As such, I do not believe a broader impact statement is required.

**Claims And Evidence:**

Yes

**Claims Explanation:**

The paper makes several claims and proves them directly in the paper. I highlight the claims made and how well they are supported below, noting particularly strong aspects of the paper:

1. Theorem 4.1 is a very general and elegant unification, subsuming prior special-case results and putting all error minimisation problems into the same categorical framework.

2. The “S-flavoured error” abstraction offers a principled way to relate error to morphisms between datasets and is a useful conceptual contribution.

3. The discussion of adjunctions as pseudo-inverses to inference gives a clear link between abstract category theory and the more intuitive notion of an “ideal” algorithm.

4. The results open the door to applying existing categorical tools such as adjoint functor theorems and colimit constructions directly to machine learning optimisation.

**Requested Changes:**

I give some explicit requested changes below, together with whether or not each change is critical for securing acceptance:

1. Please provide at least one fully worked example showing how a standard ML problem (e.g., linear or logistic regression) can be represented in this framework, including how the Kan extension arises (critical for securing acceptance).

2. Please add some discussion of computational feasibility. In the constructed 2-categories of Theorem 4.1, when is computing a left Kan extension tractable? Are there practical cases where this could be done efficiently? (critical for securing acceptance)

3. Add diagrams and informal explanations alongside key definitions to help readers build intuition. (recommended but not critical for acceptance)

4. Minor corrections for clarity and correctness (recommended but not critical):

- Break up long formal definitions with brief plain-language restatements

- Fix small typos such as “them Alg(d) is a global error minimizer” -> “then Alg(d) is a global error minimizer” (on page 7)

---

> ### Author Response · Authors · 2025-10-09
> **First response to reviewer Jc1D**
>
> Thank you very much for reviewing our paper "Learning is a Kan Extension".
>
> We were excited to see that you enjoyed the paper. To say that "This paper presents a substantial theoretical contribution" is a genuine complement that we sincerely appreciate.
>
> To address your requested changes, to the best of our knowledge we have fixed all of the small typos. We have also added two example constructions, convex optimisation and linear regression, to section four. These will hopefully make the construction of T clearer, alongside a new diagram to assist the intuitions of the reader through the construction. The example of linear regression has the added benefit of demonstrating that the Moore-Penrose psuedo inverse may be presented as an adjunction, which further solidifies the association between adjunctions, pseudo-inverses, and optimisation algorithms.
>
> To address your inquiry about computational feasabillity, we have added a discussion to the end of section four alongside a new proposition (Prop 4.4) which demonstrates how the computational complexity of the construction of Kan extensions relate to the complexity of the functions used to define the underlying error minimisation problem. In summary, the computation of a Kan extension is feasible at least whenever the computation of an optimisation algorithm is feasible.

---

> > ### Comment · Reviewer_Jc1D · 2025-10-17
> > **Thank you**
> >
> > Thank you for the response and for addressing the requested revisions. The additional examples on convex optimization and linear regression clarify the construction in Section 4, and the accompanying diagram improves readability. The discussion on computational feasibility and Proposition 4.4 also appropriately address the earlier concern regarding tractability. The revisions resolve the main points raised in the initial review.

---

### Review · Reviewer_SpYQ · 2025-08-11

**Summary Of Contributions:**

The paper recasts supervised error minimization using category theory. It defines an $S$ flavoured error as a lax 2-functor into a monoidal preorder so that information loss composes via an inequality. It then shows that left adjoints to inference yield global minimizers, independent of the chosen error, with existence ensured by adjoint functor theorems. Finally, it proves that left Kan extensions are error minimizers and that any set-theoretic error minimization problem admits a 2-categorical representation where global minimizers coincide with left Kan extensions (Theorem 4.1). This reframes algorithm design as constructing left adjoints to inference and choosing error structures from dataset morphisms.

Strengths:

S1. Establishes that all set-theoretic error minimization problems admit a 2-categorical encoding where global minimizers coincide with left Kan extensions (Theorem 4.1)—a clean bridge from ML objectives to categorical machinery.

S2. The $S$ flavoured error (lax 2-functor into a monoidal preorder) gives a principled way to model information loss and its composition via $Err(g)Err(f) \leq Err(gf)$. This both respects structure and guides error selection based on dataset morphisms.

S3. Shows adjunctions are error minimizers and connects to adjoint functor theorems to obtain sufficient conditions for optimal solutions without fixing a loss function.

Weaknesses:

W1. The paper gives representation theorem but does not demonstrate concrete instantiations on standard ML problems (e.g., explicit choices of $S,D,M,$ and computed Kan extensions), leaving practical impact to future work.

W2. The proof of Theorem 4.1 relies on building a 2-category and hom-sets engineered to select minimizers, which may feel mathematically tautological and offers limited guidance for natural problem categories.

W3. While $S$ flavoured error is elegant, how to pick $S$ and the lax 2-functor for real tasks is mostly left abstract; the requirement of a bottom element and inequality-based composition may not align cleanly with common ML losses without possible additional design choices.

**Audience:**

Yes

**Audience Explanation:**

Yes it will interest certain researchers in theoretical machine learning who value new abstractions that unify learning objectives via adjoints and Kan extensions.

**Broader Impact Concerns:**

Not very applicable here.

**Claims And Evidence:**

Yes

**Claims Explanation:**

The core category-theoretic claims are accurately and clearly supported by formal proofs within the paper’s framework. Though, the practical significance is more speculative, due to the engineered nature of the universal construction and the absence of applications to particular error minimization approaches.

**Requested Changes:**

1. Please refine the exposition, especially the introduction, and make the connections among definitions, theorems, and implications explicit. Add motivation and high-level intuition so that readers without a background in category theory can follow the results and understand their significance. The overall writing needs improvement.

2. Please add a concrete application on a standard ML task that explicitly instantiates $S,D,$ and $M$, defines dataset morphisms, and computes the left Kan extension. A brief, step by step walk-through interpreting the resulting predictor would improve accessibility and demonstrate practical utility.

3. The conclusion notes potential applications; however, developing at least one concrete instantiation would substantially enhance the paper’s appeal. I acknowledge this may be beyond scope, but in its current form the manuscript reads largely as a collection of definitions and proofs with limited guidance on practical impact.

4. Please ensure all figures are clearly labeled and explicitly referenced in the main text, with concise captions that explain their relevance to the narrative.

---

> ### Author Response · Authors · 2025-10-08
> **First response to reviewer SpYQ**
>
> Thank you very much for reviewing our paper "Learning is a Kan Extension".
>
> We really appreciate the time you have taken to read through the paper and to verify its claims.
>
> In order to address your requested changes we have added two examples, convex optimisation and linear regression, to section 4 alongside an expository diagram. These examples explicitly instantiate Err, Inf, D, and M and demonstrate how the construction of the 2-category T follows from this presentation. We hope that the inclusion of these examples will not only serve to provide clarity to the process of the construction, but also demonstrate how standard optimisation algorithms may be formatted within this framework.
>
> With regards to your request for figures to be clearly labeled and explicitly referenced in the main text, we have made sure that the figure that has been included to demonstrate the construction of T for the example of convex optimisation has been captioned and referenced within the text. For the case of the category theoretic diagrams which demostrate compositions of morphisms, it is common for diagrams of these kinds, particularly small ones, to be treated as equations following the relevant body of text. For the sake of formatting clarity and consistency with common conventions these diagrams have not been captioned. If despite this reason this change is something you still feel would benefit the paper then we will implement it.

---

### Review · Reviewer_KEQY · 2025-09-18

**Summary Of Contributions:**

- This work builds a bridge between machine learning (ML) and category theory by showing that any error minimization problem in ML can be reformulated as a Kan extension problem.
- the authors prove that given an arbitrary learning task (with a pre-defined space of models, that of data/outcomes, a prediction map, and a loss function), one can construct a categorical framework in which finding an optimal model is equivalent to computing a left Kan extension.
- In doing so, it opens the door to applying powerful categorical tools and theorems to machine learning problems.

Strength

- The paper’s main theorem is quite general – it does not just draw an analogy between one algorithm and a Kan extension, but shows every error minimization problem fits into this paradigm. This unification is a significant strength, as it consolidates many specific observations into one framework.
- By making the connection to Kan extensions precise, the work allows researchers to apply known category-theoretic results to ML. For example, a strong insight is treating loss functions as morphism-induced measures of information loss. Another insight is that a learning algorithm can be seen as a left adjoint (a kind of generalized inverse) to the data-generating process.
- The reduction of learning to adjunctions and Kan extensions is carried out in a mathematically rigorous way. For instance, Theorem 3.9 is a direct consequence of the adjunction properties and is proved by showing the required error inequality holds universally. Theorem 4.1 is constructive – the authors explicitly build the required 2-category and verify that a left Kan extension in it matches the known minimizer.


Weakness
- A notable weakness is the high level of abstraction. The paper operates in the realm of category theory (2-categories, lax 2-functors, Kan extensions), which is far from the usual toolkit of most machine learning researchers. This means the work might be challenging to digest for the typical reader in ML or even theoretical CS who isn’t already versed in category theory.
- The paper is almost entirely theoretical. While it references prior examples (e.g. anomaly detection algorithm derived via Kan extensions in separate work, and a classification algorithm in Shiebler’s work), it does not demonstrate the framework on a specific well-known learning problem within the paper.
- The main Theorem 4.1, though powerful, is somewhat non-constructive. In its proof, the authors assume the existence of a minimizer in order to construct the category $T$ – for example, they define a function $Alg: U \to M$ that picks an error-minimizing model for each dataset $d$ (in the subset $U \subseteq D$ where a minimizer exists). This means the framework encodes a solution that is already known or assumed to exist, rather than providing a method to find a solution when one is not known.
- The flip side of the paper’s generality is that the constructed 2-category in Theorem 4.1 can feel tailored or artificial to each problem. The authors essentially build $T$ in a custom way for a given learning task, encoding the error criterion into the morphisms of $T$. Some may question whether this is just a tautological embedding – after all, if you already have an optimal model, you can always embed it into a category somehow.

**Audience:**

Yes

**Audience Explanation:**

I guess only those people working on KAN will be interested.

**Broader Impact Concerns:**

Not many people will read this paper.

**Claims And Evidence:**

Yes

**Claims Explanation:**

The paper’s core results – notably Theorem 3.9 and Theorem 4.1 – are stated with clear hypotheses and conclusions, and their proofs leverage standard categorical reasoning.

**Requested Changes:**

- Include at least one worked-out example of a standard learning problem (e.g., linear regression, classification with cross-entropy loss) reformulated as a Kan extension. This would demonstrate practical relevance and clarify the constructions.
- The construction of the 2-category T is valid but quite dense and hard to follow. Currently, the argument is heavy with logical case-splitting.
- The paper frames all learning problems as Kan extensions, but does not sufficiently acknowledge the limitations (e.g., non-existence of minimizers, the somewhat ad hoc construction of T).

---

> ### Author Response · Authors · 2025-10-08
> **First response to reviewer KEQY**
>
> Thank you very much for reviewing our paper "Learning is a Kan Extension".
>
> It was gratifying to see that you appreciated the generality of the construction. Its common to colloquially refer to Kan extensions as a kind of error minimiser, but to our knowledge, there has not previously been a result which presents this notion rigorously. Hopefully, these results can provide a unification for the constuctions that future researchers explore, as well as the s-flavoured error providing a methodology for translating the more abstract presentations into the more familiar setting of set theoretic error.
>
> Unfortunately, as you identified, the high level of abstraction makes portions of the work difficult to digest. Particularly the construction of the 2-category T. In an attempt to remedy this issue, we have added two example constructions, convex optimisation and linear regression, to section four. Hopefully, these examples will both clarify the mechanism through which 4.1 operates as well demonstrating how one can present traditional optimisation problems within this framework. These examples also have the benefit of demonstrating that the Moore-Penrose pseudo inverse can be viewed as an adjunction, further solidifying the association between adjunctions, pseudo inverses, and optimisation algorithms.
>
> You mentioned one further requested change in the acknowledgement of limitations, particularly in the non-existence of minimisers. A discussion on the effect of the existence or lack thereof of global error minimisors is present after Corollary 3.12 and also in the proof of theorem 4.1. It is an important constraint because it is one of the primary reasons for generalising the results beyond adjunctions to Kan extensions. The presentation of an algorithm as an adjunction can only exist if all error minimisors exist (such as in the case of linear regression). However, when only some error minimisors exist then the results may be presented via Kan extensions, as one minimiser exists per extension. If no error minimisors exist then no Kan extensions can exist. I hope this suitably clarifies the point you identified.

---

### Decision · Action_Editor_uu6L · 2025-11-01

**Recommendation:** Accept with minor revision

**Additional Comments:**

Overall, the reviewers had positive comments for this paper, and I am essentially in agreement with them. Two of the reviewers voted for acceptance, and one was leaning to reject, but all reviewers felt the paper had strong clarity and solid claims, as well as a potential audience in the TMLR community. While concerns were raised about how large this potential audience will be, I think it is satisfactory for acceptance to TMLR.

I have recommended "acceptance with minor revision". The revision I have in mind is with respect to formatting; are the authors using the TMLR style file correctly? The text looks strange to me, I think it is the spacing, but I cannot pin it down exactly. The fonts are the same, I believe, but the spacing between letters is much tighter in this submission than in any other typical submission. Please check that the paper is made strictly according to the TMLR style guidelines.

**Audience:**

Yes

**Audience Explanation:**

This is a tricky one, since the content is extremely abstract, and at present I think for most readers it is difficult to imagine how the abstract insights available in the literature for computing Kan extensions, however "efficient" they may be, could translate to fruitful, concrete machine learning algorithms. That said, establishing connections between different research domains is assuredly a worthwhile pursuit, and connections with high generality mean that a wide range of learning tasks can potentially be captured. Combine this with the concise but well-written prose, I think it is safe to say that at least some individuals in the TMLR audience would be interested, and could gain something from the findings in this work.

**Claims And Evidence:**

Yes

**Claims Explanation:**

In this work, the authors show how certain mathematical operations, which can be naturally interpreted within the context of machine learning, also can be understood in the language of category theory. Their formulation is quite general; essentially a machine learning model, once specified, is seen as determining a function which can be applied to a set of input data. The resulting set of output data can then be compared with alternatives and quantitatively evaluated (assigned a real number penalty). In this setup, when a certain kind of optimality is obtained (their "error minimizers"), a clear link to "Kan extensions" is made.

The content is quite abstract, but the exposition is clear, precise, and well-written. All the reviewers were in agreement on this point.

---

> ### Author Response · Authors · 2025-11-10
> **First Response to Decision**
>
> Thank you for accepting our paper and for your positive comments. We think we have fixed the formatting issue that you identified, and have submitted the camera ready version. Let us know if our changes are satisfactory.